# Early unfractionated heparin treatment in patients with STEMI – trial design and rationale

**Misa Fister** [1,2]*, **Ursa Mikuz**[1], **Klemen Ziberna**[2], **Danilo Franco** [3], **Peter Radsel**[1,2], **Matjaz Bunc**[1,2], **Marko Noc**[1,2], **Tomaz Goslar** [1,2]

**1** Department of intensive internal medicine, University Medical Center Ljubljana, Ljubljana, Slovenia,
**2** Medical faculty, University of Ljubljana, Ljubljana, Slovenia, **3** Department of Advanced Biomedical Science, University of Naples "Federico II", Naples, Italy

* misa.fister@kclj.si

**Data Availability Statement:** No datasets were generated or analysed during the current study. All relevant data from this study will be made available upon study completion.

## Abstract

The early unfractionated heparin (UFH) treatment in patients with ST-elevation myocardial infarction (STEMI) is a single-center, open-label, randomized controlled trial. The study population are patients with STEMI that undergo primary percutaneous coronary intervention (PPCI). The trial was designed to investigate whether early administration of unfractionated heparin immediately after diagnosis of STEMI is beneficial in terms of patency of infarct-related coronary artery (IRA) when compared to established UFH administration at the time of coronary intervention. The patients will be randomized in 1:1 fashion in one of the two groups. The primary efficacy endpoint of the study is Thrombolysis in myocardial infarction (TIMI) flow grades 2 and 3 on diagnostic coronary angiography. Secondary outcome measures are: TIMI flow after PPCI, progression to cardiogenic shock, 30-day mortality, ST-segment resolution, highest Troponin I and Troponin I values at 24 hours. The safety outcome is bleeding complications. The study of early heparin administration in patients with STEMI will address whether pretreatment with UFH can increase the rate of spontaneous reperfusion of infarct-related coronary artery.

## Introduction

Ischemic heart disease is the leading cause of death worldwide, accounting for 20% of all deaths in Europe. STEMI accounts for considerable part of cardiovascular disease morbidity and mortality with incidence in Europe ranging from 43–144/100,000 per year and from 50/100,000 per year in the US. Mortality at 1 year in STEMI patients treated with coronary angiography is reported at approximately 10%. Immediate PPCI and opening of occluded coronary artery is the most important intervention that reduces STEMI-related mortality and morbidity [1].

Guidelines for the management of patients with STEMI only recommend unfractionated heparin (UFH) at the time of PPCI, and the data on early treatment with UFH at the time of diagnosis are scarce [1, 2]. Contrary to the international guidelines, our center has been using

**Funding:** The author(s) received no specific funding for this work.

**Competing interests:** The authors have declared that no competing interests exist.

UFH at the dose of 70–100 IE/kg as a pretreatment immediately once the diagnosis of STEMI and the decision for PPCI was made.

In Slovenia, a country with around 2,000,000 inhabitants, STEMI patients are referred to one of two 24/7 catheterization laboratories. University Medical Centre of Ljubljana is one of them and covers around 1,200,000 inhabitants. If a doctor in the emergency room suspects myocardial infarction consultation with an intensive care physician/cardiologist is made. Based on the electrocardiogram and the clinical picture, the cardiologist decides whether to refer the patient directly to a cath lab or to the nearest emergency room. The consulting doctor will also advise on the medication to be used. Patients referred directly to the cath lab will be given aspirin 250/500 mg po and approximately 100 units/kg UFH. Pre-treatment with P2Y12 is not administered.

In the recent COOL AMI EU Pivotal trial and the COOL AMI EU Pilot, the multicenter, prospective randomized controlled trials, we noticed a higher proportion of spontaneous reperfusions of culprit artery in our patients, receiving early UFH in the field when compared to other centers without early UFH treatment [3, 4].

In the proposed trial, we intend to investigate effectiveness and safety of early UFH administration compared to late administration at the time of coronary intervention.

## Materials and methods

### Participants, interventions and outcomes

**Study objectives.** Early unfractionated heparin treatment in patients with STEMI is a single-center, open-label, randomized controlled trial and will determine if early administration of UFH in the field at the time of established diagnosis and decision for PPCI is superior to the recommended application at the time of PPCI.

The trial is registered under www.clinicaltials.gov: NCT05247424.

**Patient population.** Study population will consist of 600 STEMI patients referred for PPCI at University Medical Centre of Ljubljana, Slovenia. Flowchart of the study is presented in Fig 1. Case report form is included as S1 Checklist.

*Inclusion criteria*. Inclusion criteria are based on standard definition of STEMI (at least two contiguous leads with ST-segment elevation $\geq 2.5$ mm in men $< 40$ years, $\geq 2$ mm in men $\geq 40$ years, or $\geq 1.5$ mm in women in leads $V_2$–$V_3$ and/or $\geq 1$ mm in the other leads in the absence of left ventricular hypertrophy or left bundle branch block) [1], with additional limitation for duration of symptoms.

Inclusion criteria are:

- Adult patients with ST-elevation acute myocardial infarction

- Chest pain for less than 6 hours

  *Exclusion criteria*. Exclusion criteria are:

- Cardiac arrest without regaining consciousness

- Chest pain for more than 6 hours

- Hemodynamic impairment (cardiogenic shock)

- Pregnant women

An overview of the timepoints for assessment of study endpoints

| | STUDY PERIOD | | | | | | | |
|---|---|---|---|---|---|---|---|---|
| | Enrolment | Allocation | Post allocation | | | | | Close-out |
| | Presentation (emergency department) | Presentation (emergency department | Cath lab arrival | Coronary angiography | End of coronary angiography | Hospital stay | Hospital discharge | 30 days |
| Timepoint | $-t_1$ | 0 | $t_1$ | $t_2$ | $t_3$ | $t_4$ | $t_5$ | $t_6$ |
| ENROLMENT: | | | | | | | | |
| Eligibility screen | x | | | | | | | |
| Informed consent | | | ———————————————————— | | | | | |
| Allocation | | x | | | | | | |
| INTERVENTIONS: | | | | | | | | |
| Coronary angiography | | | | x | | | | |
| Heparin in the field | | x | | x | | | | |
| Heparin in the cath lab | | | | x | | | | |
| ASSESMENTS: | | | | | | | | |
| Electrocardiogram | x | | | | x | | | |
| TIMI flow | | | | x | x | | | |
| Troponin | | | | | | ———————————— | | |
| Bleeding | | | ———————————————————— | | | | | |
| Cardiogenic shock | | | ———————————————————— | | | | | |
| Vital status | | | ———————————————————————————————— | | | | | |

**Fig 1. An overview of the timepoints for assessment of study endpoints.**

## Assignment of interventions

Patient randomization is performed directly after establishing diagnosis of STEMI and decision for PPCI has been made. Randomization is in a 1:1 ratio to:

1. Immediate application of UFH intravenously at a dose of 70–100 units/kg on top of standard treatment. Additional UFH is added after diagnostic coronary angiography according to activated clotting time (ACT) when PPCI is performed.

2. Standard treatment and application of intravenous UFH only after diagnostic coronary angiography at a recommended dose of 70–100 units/kg when PPCI is performed.

When the ER doctor calls the intensivist about an eligible STEMI patient, randomization is performed using random permutated blocks via secure online randomization service (www.sealedenvelope.com). The website can be opened on a smartphone or a computer. After entering the password for the study, the randomizing physician's email is entered, followed by the unique patient ID. The inclusion and exclusion criteria are then checked and the "Randomize" field is clicked. The result of the randomization to group A–heparin on the field - or B–no heparin - is displayed on the screen and sent to the randomizing physician's email. The doctor performing the randomization then informs the ER doctor about the therapy and dosage. We started randomizing patients on March 10th 2022. Predicted end of the study is 31st March 2025.

Upon arrival to the cath lab, immediate coronary angiography is performed via radial or femoral approach at the discretion of the interventional radiologist. ACT is measured in all patients who receive pretreatment with UFH and when PPCI is needed, an additional bolus of UFH is given for a target ACT of 250–350 seconds. In patients without pretreatment, UFH is administered according to guidelines [1] in the dose of 70–100 units/kg when PPCI is needed. Additional anticoagulation or glycoprotein IIb/IIIa (eptifibatide) can be administered at the discretion of the interventional cardiologist or an additional bolus of UFH is added in long procedures.

The vascular access site is closed with a closure device or manual compression. The patient receives a P2Y12 inhibitor at the discretion of the interventional cardiologist immediately after PPCI while still in the cath lab.

Patients are admitted to cardiac intensive care unit usually for a period of 24 hours; there, serial troponin I measurements are performed. Possible bleeding complications are prospectively monitored and recorded.

## Outcome definitions

**Primary outcome.**   The primary efficacy endpoint of the study is TIMI [5] flow grade 2 or 3 on diagnostic coronary angiography.

**Secondary outcomes.**   The secondary endpoints aim to assess infarct size, effectiveness of reperfusion, hemodynamic deterioration and survival [6–8]. Cardiogenic shock will be defined clinically (systolic blood pressure of less than 90 mmHg for longer than 30 minutes or the use of catecholamine therapy to maintain systolic pressure of at least 90 mmHg [8].

Secondary endpoints are:

- Final TIMI flow grades 2 and 3

- Highest Troponin I value

- Troponin I value 24 hours after PPCI

- ST-segment resolution in a single lead with maximum baseline ST-segment elevation

- Progression to cardiogenic shock

- 30-day mortality

## Safety endpoint

Safety endpoint will be the occurrence of bleeding according to the Bleeding Academic Research Consortium (BARC) definition 3–5 [9]. We do not expect serious bleeding complications in STEMI patients. A potential harm of heparin in the prehospital setting would be the case of misdiagnosis of STEMI, e.g. intracerebral hemorrhage with ECG changes or aortic dissection mimicking STEMI by impairing coronary flow. These complications are also closely monitored.

## Data collection, management, and analysis

**Sample size calculation.**   Several published observational studies have investigated pretreatment with UFH [10–13]. The reported difference in rates of spontaneous reperfusion varies and depends on the time between UFH administration and PPCI.

In a propensity-matched study of 552 matched patients, Bloom et al. reported a significantly lower proportion of patients with a TIMI 0 or 1 flow in the IRA in those who received UFH in fixed bolus doses of 4000 units and 1000 units at hourly intervals during transport prehospital (66% vs. 76%, p<0.001) compared with those who did not [11]. Investigators in the observational substudy of the TASTE trial reported a lower incidence of TIMI 0 or 1 in patients who received an average of 5000 units of UFH pre-hospital or in the emergency department (73.1% vs. 80.9%, p<0.001) compared to those who did not [12]. Another observational study by Giralt et al. also showed that pretreatment with UFH in a fixed intravenous dose resulted in a lower rate of TIMI 0 or 1 (69.7% vs. 78.8%, p<0.001) compared to the post-treatment group.

In addition, the time-dependent effect of UFH administration was evident with higher rates of spontaneous reperfusion (TIMI 2 or 3) with shorter duration of administration from symptom onset [13].

The sample size calculation for this study is based on the historical baseline rate of TIMI 0 or 1 in STEMI patients arriving at our clinical center and the estimated benefit of prehospital UFH administration from the previously mentioned studies. The historical baseline rate of TIMI 0 or 1 flow at the University Medical Centre of Ljubljana is 58.5% with UFH pretreatment. We designed the study to detect a difference of 11.5% in the experimental group, which corresponds to 70% of patients with TIMI 0 or 1. To achieve a power of 80% for detecting this difference at a significance level of 5%, a total of 538 patients are required. In addition, we considered a drop-out rate of 2.5%, which increases the total sample size required to 598. Sample size was calculated using the Sealed Envelope Ltd. 2012 Power Calculator for Binary Outcome Superiority Trial Web Application (https://www.sealedenvelope.com/power/binary-superiority/).

**Data analysis.** All data will be analyzed according to the intention-to-treat principle. Fisher's exact test will be used to compare TIMI 2 or 3 flow rates in both arms. An odds ratio with a 95% confidence interval will be calculated as an estimator of early UFH administration.

For all secondary endpoints, the effect of early UFH administration with a corresponding 95% confidence interval will be estimated.

We assume there will be very little missing data for primary endpoint analysis. Missing data for secondary endpoints will be replaced by imputation of missing data.

Pre-defined subgroup analysis will be performed for sex, age (<65 years vs. >65 years), culprit artery (left anterior descending, left circumflex, right and side branch), time from onset of symptoms to heparin administration (<2 hours, 2–4 hours, >4 hours), and time from heparin administration to diagnostic angiography (<1 hour, >1 hour).

## Ethics and dissemination

### Research ethics approval

The study was approved by the Ethics Committee of Republic of Slovenia (0120-591/2021/3) on January 20th 2022. Since pretreatment with UFH is a standard of care in Slovenia and due to the nature of the emergency situation in the case of STEMI, Ethics Committee granted us a waiver to obtain informed consent at the time of randomization. Informed consent will be obtained at the time of PPCI or later during hospitalization, depending on mental condition of the patient. In case of death, Ethics Committee granted approval of data utilization without patient consent.

## Discussion

Current guidelines on treatment of STEMI recommend UFH as one of anticoagulant agents in patients undergoing PPCI, but the exact timing is not defined [1, 2]. Multiple observational studies and subgroup analyses have reported beneficial effects of pretreatment with UFH on the rate of spontaneous reperfusion of IRA [10–13]. To our knowledge, this is the first randomized trial investigating effects of pretreatment with UFH.

The duration of coronary thrombosis is likely to have a significant influence on effectiveness of UFH on the patency of IRA as indicated in one observational study [13]. With a longer duration of coronary thrombosis, the composition and organization of a thrombus are changing, making it less susceptible to endogenous fibrinolysis and also making UFH less effective. To show effectiveness of early UFH administration, we limited the patient population only to those with the duration of symptoms at less than 6 hours. We decided against any age limitation for our inclusion criteria to mimic the real patient population. However, we decided to

exclude patients who remain unconscious after cardiac arrest and those in cardiogenic shock at the time of randomization. These patients represent a different subpopulation and would at this point only complicate the interpretation of results.

Side effects of early UFH administration are unlikely since it is an established practice and so far, we did not observe any higher rates of bleeding.

The trial is not blinded. Distribution of placebo or heparin to all prehospital units and education of all emergency physicians would be too complicated and costly. However, we tried to reduce chances of bias as much as possible. We decided for central computer-based randomization by consulting cardiologist/intensivist. An experienced interventional cardiologist who reviews and evaluates patient TIMI flow is not involved in patient care and will also be blinded to patient assignment. The culprit artery will be determined based on the ECG and angiographic features at the first coronary angiography. An acute culprit lesion was defined as abrupt occlusion or TIMI 2/3 flow with angiographic images suggestive of thrombus or ulcerated plaque [14].

Bleeding is assessed by the nurses in the intensive care unit and recorded in an established protocol for monitoring patients after coronary angiography.

The study started recruiting patients on 10[th] of March 2022. 56 patients were randomized by the end of May. With the rate of about 15–20 patients per month, the study should be completed by the spring of 2025.

## Conclusions

Multiple subgroup analyses or observational studies have indicated potential benefit of pretreatment with UFH in patients with STEMI referred for PPCI. However, prospective randomized data to prove the benefit of this simple and inexpensive intervention in reducing the rate of spontaneous reperfusion of infarct-related coronary artery or possibly even mortality is still missing.

This trial was designed to test the hypothesis that pretreatment with UFH in patients with STEMI increases the rate of spontaneous reperfusion compared to standard administration at the time of PPCI.

## Supporting information

**S1 Checklist. SPIRIT 2013 checklist: Recommended items to address in a clinical trial protocol and related documents\*.**
(DOC)

## Author Contributions

**Conceptualization:** Misa Fister, Ursa Mikuz.

**Data curation:** Misa Fister, Ursa Mikuz, Peter Radsel, Matjaz Bunc, Tomaz Goslar.

**Formal analysis:** Klemen Ziberna, Danilo Franco, Peter Radsel, Marko Noc, Tomaz Goslar.

**Funding acquisition:** Tomaz Goslar.

**Methodology:** Klemen Ziberna, Marko Noc.

**Project administration:** Misa Fister, Tomaz Goslar.

**Supervision:** Misa Fister.

**Writing – original draft:** Tomaz Goslar.

**Writing – review & editing:** Misa Fister.

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
