## [Decision Letter · Decision Letter 0]

27 Feb 2024

PONE-D-24-00399Early unfractionated heparin treatment in patients with STEMI – trial design and rationalePLOS ONE

Dear Dr. Fister,

Thank you for submitting your manuscript to PLOS ONE. After careful consideration, we feel that it has merit but does not fully meet PLOS ONE’s publication criteria as it currently stands. Therefore, we invite you to submit a revised version of the manuscript that addresses the points raised during the review process.

**ACADEMIC EDITOR: Please address the comments given by the reviewer 2, reviewer 3 and reviewer 4.**

We look forward to receiving your revised manuscript.

Kind regards,

Pasyodun Koralage Buddhika Mahesh

Academic Editor

PLOS ONE

3. PLOS requires an ORCID iD for the corresponding author in Editorial Manager on papers submitted after December 6th, 2016. Please ensure that you have an ORCID iD and that it is validated in Editorial Manager. To do this, go to ‘Update my Information’ (in the upper left-hand corner of the main menu), and click on the Fetch/Validate link next to the ORCID field. This will take you to the ORCID site and allow you to create a new iD or authenticate a pre-existing iD in Editorial Manager. Please see the following video for instructions on linking an ORCID iD to your Editorial Manager account: https://www.youtube.com/watch?v=_xcclfuvtxQ.

4. Please ensure that you refer to Figure 1 in your text as, if accepted, production will need this reference to link the reader to the figure.

Reviewers' comments:

Reviewer's Responses to Questions

**Comments to the Author**

1. Does the manuscript provide a valid rationale for the proposed study, with clearly identified and justified research questions?

Reviewer #1: Yes

Reviewer #2: Yes

Reviewer #3: Yes

Reviewer #4: Yes

2. Is the protocol technically sound and planned in a manner that will lead to a meaningful outcome and allow testing the stated hypotheses?

Reviewer #1: Yes

Reviewer #2: Yes

Reviewer #3: Yes

Reviewer #4: Yes

3. Is the methodology feasible and described in sufficient detail to allow the work to be replicable?

Reviewer #1: Yes

Reviewer #2: Yes

Reviewer #3: Yes

Reviewer #4: Yes

4. Have the authors described where all data underlying the findings will be made available when the study is complete?

Reviewer #1: Yes

Reviewer #2: Yes

Reviewer #3: Yes

Reviewer #4: Yes

5. Is the manuscript presented in an intelligible fashion and written in standard English?

Reviewer #1: Yes

Reviewer #2: Yes

Reviewer #3: Yes

Reviewer #4: Yes

6. Review Comments to the Author

You may also provide optional suggestions and comments to authors that they might find helpful in planning their study.

Reviewer #1: This study is going to answer an important question in the area of STEMI and primary PCI. It’s well conducted and waiting to see the findings.

Reviewer #2: The authors have described the trial design and rationale of early unfractionated heparin treatment in patients with STEMI. The following comments are given with the hope that these will be beneficial to them:

1. The authors have annexed the SPIRIT checklist and have included most of the items. It is suggested to use the subheadings according to the SPIRIT checklist as shown below:

-Administrative information

-Introduction

-Methods: Participants, interventions, and outcomes

-Methods: Assignment of interventions (for controlled trials)

-Methods: Data collection, management, and analysis

-Methods: Monitoring

-Ethics and dissemination

-Appendices

2. Suggest including country-specific data as well as center-specific data in the introduction

3. Please improve the clarity of the section on sample size calculation in the methods

4. Suggest including bit more details on safety considerations

Reviewer #3: The authors have prepared the study protocol on “Early unfractionated heparin treatment in patients with STEMI – trial design and rationale”. This study will provide important evidence. The protocol in general has been well written. I have made the following comments to get more clarity of the study.

1. Please elaborate the randomization process a bit more.

2. Please state how investigators would ensure blinding of assessors.

3. Please mention whether the time period from the admission to the start of intervention at the catheter laboratory is captured as a variable since it could be a confounder for the outcome of the study

Reviewer #4: I would prefer if authors mention about the rationale for the exclusion criteria. Please describe more about the analysis in relation to the objectives of this protocol.

7. PLOS authors have the option to publish the peer review history of their article (what does this mean?). If published, this will include your full peer review and any attached files.

Reviewer #1: **Yes: **Aruna Wijesinghe

Reviewer #2: **Yes: **I.O.K.K.Nanayakkara

Reviewer #3: **Yes: **Vidura Jayasinghe

Reviewer #4: No

---

## [Author Response · Author response to Decision Letter 0]

3 Apr 2024

Response to reviewers

Thank you for your helpful remarks and a chance to improve our manuscript. Separate answers to reviewers are discussed below. 

Reviewer #1: This study is going to answer an important question in the area of STEMI and primary PCI. It’s well conducted and waiting to see the findings.

Answer: We thank reviewer for his positive comment.

Reviewer #2: The authors have described the trial design and rationale of early unfractionated heparin treatment in patients with STEMI. The following comments are given with the hope that these will be beneficial to them:

1. The authors have annexed the SPIRIT checklist and have included most of the items. It is suggested to use the subheadings according to the SPIRIT checklist as shown below:

-Administrative information

-Introduction

-Methods: Participants, interventions, and outcomes

-Methods: Assignment of interventions (for controlled trials)

-Methods: Data collection, management, and analysis

-Methods: Monitoring

-Ethics and dissemination

-Appendices

Answer: We have rearranged the manuscript according to PlosOne guidelines and Spirit Checklist. 

2. Suggest including country-specific data as well as center-specific data in the introduction

Answer: We have moved the country-specific and center-specific data to the introduction section. The following explanation has been added:

In Slovenia, a country with around 2,000,000 inhabitants, STEMI patients are referred to one of two 24/7 catheterization laboratories. University Medical Centre of Ljubljana is one of them and covers around 1,200,000 inhabitants. If a doctor in the emergency room suspects myocardial infarction consultation with an intensive care physician/cardiologist is made. Based on the electrocardiogram and the clinical picture, the cardiologist decides whether to refer the patient directly to a cath lab or to the nearest emergency room. The consulting doctor will also advise on the medication to be used. Patients referred directly to the cath lab will be given aspirin 250/500 mg po and approximately 100 units/kg UFH. Pre-treatment with P2Y12 is not administered.

3. Please improve the clarity of the section on sample size calculation in the methods

Answer: The sample size calculation in the methods section has been expanded and rewritten in the following way:

Several published observational studies have investigated pretreatment with UFH [10–13]. The reported difference in rates of spontaneous reperfusion varies and depends on the time between UFH administration and PPCI.

In a propensity-matched study of 552 matched patients, Bloom et al. reported a significantly lower proportion of patients with a TIMI 0 or 1 flow in the IRA in those who received UFH in fixed bolus doses of 4000 units and 1000 units at hourly intervals during transport prehospital (66% vs. 76%, p<0.001) compared with those who did not [11]. Investigators in the observational substudy of the TASTE trial reported a lower incidence of TIMI 0 or 1 in patients who received an average of 5000 units of UFH pre-hospital or in the emergency department (73.1% vs. 80.9%, p<0.001) compared to those who did not [12]. Another observational study by Giralt et al. also showed that pretreatment with UFH in a fixed intravenous dose resulted in a lower rate of TIMI 0 or 1 (69.7% vs. 78.8%, p<0.001) compared to the post-treatment group. In addition, the time-dependent effect of UFH administration was evident with higher rates of spontaneous reperfusion (TIMI 2 or 3) with shorter duration of administration from symptom onset [13].

The sample size calculation for this study is based on the historical baseline rate of TIMI 0 or 1 in STEMI patients arriving at our clinical center and the estimated benefit of prehospital UFH administration from the previously mentioned studies. The historical baseline rate of TIMI 0 or 1 flow at the University Medical Centre of Ljubljana is 58.5% with UFH pretreatment. We designed the study to detect a difference of 11.5% in the experimental group, which corresponds to 70% of patients with TIMI 0 or 1. To achieve a power of 80% for detecting this difference at a significance level of 5%, a total of 538 patients are required. In addition, we considered a dropout rate of 2.5%, which increases the total sample size required to 598. Sample size was calculated using the Sealed Envelope Ltd. 2012 Power Calculator for Binary Outcome Superiority Trial Web Application (https://www.sealedenvelope.com/power/binary-superiority/).

4. Suggest including bit more details on safety considerations

Answer: We have added a paragraph on safety considerations. They are highlighted on the “Safety endpoint” section. The following paragraph was added:

We do not expect serious bleeding complications in STEMI patients. A potential harm of heparin in the prehospital setting would be the case of misdiagnosis of STEMI, e.g. intracerebral hemorrhage with ECG changes or aortic dissection mimicking STEMI by impairing coronary flow. These complications are also closely monitored.

Reviewer #3: The authors have prepared the study protocol on “Early unfractionated heparin treatment in patients with STEMI – trial design and rationale”. This study will provide important evidence. The protocol in general has been well written. I have made the following comments to get more clarity of the study.

1. Please elaborate the randomization process a bit more.

Answer: The randomization process has been explained, and is highlighted in the “Assignment of interventions” section. 

When the ER doctor calls the intensivist about an eligible STEMI patient, randomization is performed using random permutated blocks via secure online randomization service (www.sealedenvelope.com). The website can be opened on a smartphone or a computer. After entering the password for the study, the randomizing physician's email is entered, followed by the unique patient ID. The inclusion and exclusion criteria are then checked and the “Randomize" field is clicked. The result of the randomization to group A – heparin on the field - or B – no heparin - is displayed on the screen and sent to the randomizing physician's email. The doctor performing the randomization then informs the ER doctor about the therapy and dosage.

2. Please state how investigators would ensure blinding of assessors.

Answer: The blinding of the assessors is further explained in the “Discussion” section, the paragraph added is highlighted:. 

The trial is not blinded. Distribution of placebo or heparin to all prehospital units and education of all emergency physicians would be too complicated and costly. However, we tried to reduce chances of bias as much as possible. We decided for central computer-based randomization by consulting cardiologist/intensivist. An experienced interventional cardiologist who reviews and evaluates patient TIMI flow is not involved in patient care and will also be blinded to patient assignment. The culprit artery will be determined based on the ECG and angiographic features at the first coronary angiography. An acute culprit lesion was defined as abrupt occlusion or TIMI 2/3 flow with angiographic images suggestive of thrombus or ulcerated plaque.

3. Please mention whether the time period from the admission to the start of intervention at the catheter laboratory is captured as a variable since it could be a confounder for the outcome of the study

Answer: We are collecting several times, including time of the onset of pain, first medical contact, time of heparin addministration, time to arrival to the cathlab, and time of the start and end of coronary angiography. All collected data can be seen in the attached CRF (Case Report Form) as supporting information we added to the paper. 

Reviewer #4: 

1. I would prefer if authors mention about the rationale for the exclusion criteria. 

Answer: We apply exclusion criteria to eliminate possible confounders with the primary efficacy and safety endpoints. The reason for the time limitation of the included patients was a reference to previous reports of a possible time-dependent effect of UFH, with a higher probability of a positive effect at an earlier time point. To prove the beneficial effect of early UFH administration, we decided to limit the time to 6 hours. However, a pre-planned subgroup analysis is planned to determine the effect of UFH on the age of the thrombus. Unconscious OHCA and cardiogenic shock patients represent a different subgroup of patients and were therefore excluded.

2. Please describe more about the analysis in relation to the objectives of this protocol.

Analysis of the objectives of the protocol has been now more thoroughly described under »Discussion«. The following paragraph was added: 

An experienced interventional cardiologist who reviews and evaluates patient TIMI flow is not involved in patient care and will also be blinded to patient assignment. The culprit artery will be determined based on the ECG and angiographic features at the first coronary angiography. An acute culprit lesion was defined as abrupt occlusion or TIMI 2/3 flow with angiographic images suggestive of thrombus or ulcerated plaque.

---

## [Decision Letter · Decision Letter 1]

24 Apr 2024

Early unfractionated heparin treatment in patients with STEMI – trial design and rationale

PONE-D-24-00399R1

Dear Dr. Fister,

We’re pleased to inform you that your manuscript has been judged scientifically suitable for publication and will be formally accepted for publication once it meets all outstanding technical requirements.

Kind regards,

Pasyodun Koralage Buddhika Mahesh

Academic Editor

PLOS ONE

Additional Editor Comments (optional):

Based on the comments of the reviewers and my own observations, it is concluded that the authors have addressed all previous comments satisfactorily. The additional suggestion given by Reviewer-2 is be considered by the authors when a manuscript is written following the completion of the research study.

Reviewers' comments:

Reviewer's Responses to Questions

**Comments to the Author**

1. Does the manuscript provide a valid rationale for the proposed study, with clearly identified and justified research questions?

Reviewer #2: Yes

Reviewer #3: Yes

Reviewer #4: Yes

2. Is the protocol technically sound and planned in a manner that will lead to a meaningful outcome and allow testing the stated hypotheses?

Reviewer #2: Yes

Reviewer #3: Yes

Reviewer #4: Yes

3. Is the methodology feasible and described in sufficient detail to allow the work to be replicable?

Reviewer #2: Yes

Reviewer #3: Yes

Reviewer #4: Yes

4. Have the authors described where all data underlying the findings will be made available when the study is complete?

Reviewer #2: Yes

Reviewer #3: Yes

Reviewer #4: Yes

5. Is the manuscript presented in an intelligible fashion and written in standard English?

Reviewer #2: Yes

Reviewer #3: Yes

Reviewer #4: Yes

6. Review Comments to the Author

You may also provide optional suggestions and comments to authors that they might find helpful in planning their study.

Reviewer #2: Its better if more details are needed in the methods section on how the data analyses were done. As an example how were verification done (e.g. whether two investigators analysed the data independently etc.)

Reviewer #3: the manuscript has been accepted the revised manuscript for the publication as authors have answered my concerns.

Reviewer #4: All criteria required for publication is adequately attended. Statistical analysis is described adequately.

7. PLOS authors have the option to publish the peer review history of their article (what does this mean?). If published, this will include your full peer review and any attached files.

Reviewer #2: **Yes: **I.O.K.K.Nanayakkara

Reviewer #3: **Yes: **Vidura Jayasinghe

Reviewer #4: No

---

## [Editor Report · Acceptance letter]

29 Apr 2024

PONE-D-24-00399R1 

PLOS ONE

Dear Dr. Fister, 

I'm pleased to inform you that your manuscript has been deemed suitable for publication in PLOS ONE. Congratulations! Your manuscript is now being handed over to our production team.

Kind regards, 

on behalf of

Dr. Pasyodun Koralage Buddhika Mahesh 

Academic Editor

PLOS ONE